# MA³S: Model-Agnostic Active Annotation Strategy for Crowdsourcing

**Wenjun Zhang** [1] **Liangxiao Jiang** [2] **Chaoqun Li** [3] **Shanshan Si** [2]

## Abstract

In crowdsourcing scenarios, to mitigate the impact of noisy labels assigned by non-expert workers, each instance is typically annotated multiple times by different workers. However, repeated annotation can introduce instance- or label-level redundancy, thereby inflating annotation costs. Despite its practical importance, research on repeated annotation strategies remains limited, and no existing strategy simultaneously avoids being offline, instance-unaware, and model-centric. In this paper, we propose a model-agnostic active annotation strategy, MA³S, that addresses these limitations: (1) To reduce label redundancy caused by offline procedure, MA³S estimates instance uncertainties with a margin-Beta distribution and updates them online as new labels arrive. (2) To prevent instance redundancy induced by instance-unaware designs, MA³S constructs a nearest-neighbor graph to propagate instance uncertainties, reducing repeated annotations of similar instances. (3) To avoid being model-centric, MA³S actively selects instances for annotation based solely on the estimated uncertainties, without relying on model feedback. Extensive experiments on synthetic and real-world datasets demonstrate that MA³S consistently outperforms existing annotation strategies.

## 1. Introduction

Crowdsourcing reduces data annotation costs by employing imperfect crowd workers in place of domain experts (Zhang et al., 2025c; Meir et al., 2024), which is commonly used to collect large-scale training data for supervised learning (Zhang et al., 2021; Li et al., 2025). Compared with domain experts, crowd workers are more affordable and easier to recruit, yet their limited expertise can introduce noisy labels (Li et al., 2021; Jiang et al., 2022; Kou et al., 2025a). To mitigate the impact of these noisy labels, repeated annotation is widely adopted, whereby each instance is annotated multiple times by different workers (Sheng et al., 2008). While this approach effectively offsets random noise, it still exhibits limitations, including repeatedly annotating similar instances (instance redundancy) and over-annotating easy instances (label redundancy) (Zhang, 2022; Chen et al., 2024). To prevent such redundancy from inflating annotation costs, designing more effective repeated annotation strategies warrants greater attention.

The core of designing a repeated annotation strategy is to iteratively identify the instance with the highest benefit for annotation, i.e., instance selection. Unlike worker selection in task assignment (Wijenayake et al., 2023; Zhao et al., 2022), instance selection need not assume that worker abilities are known, which better reflects practical crowdsourcing scenarios. Currently, research explicitly focused on repeated annotation strategies remains limited, so we also review work on coreset (Zhang et al., 2025a; Maalouf et al., 2024; Bardenet et al., 2024) and active learning from crowds (Li et al., 2015; Zhang et al., 2024; Kou et al., 2025b), both of which can be adapted into annotation strategies. All these strategies we survey can be broadly categorized as model-agnostic or model-centric. Specifically, model-agnostic strategies do not rely on predictive model feedback and select instances via uniform allocation, uncertainty estimation, or coreset construction. Model-centric strategies train a model on already annotated instances and then select subsequent instances based on model feedback.

Despite the progress achieved by the above strategies, noteworthy limitations still remain. We discuss these limitations in three aspects: the offline procedure, instance-unaware design, and the model-centric paradigm. Specifically, coreset-based strategies select a representative subset from the entire dataset and then annotate the subset in an offline procedure. Such offline procedure cannot exploit information incorporated during annotation, which can lead to label redundancy. Uncertainty-based strategies estimate uncertainties via entropy or Bayesian statistics but do not utilize instance-specific information. Consequently, they are

[1]School of Information Engineering, Zhongnan University of Economics and Law, Wuhan 430073, China [2]School of Computer Science, China University of Geosciences, Wuhan 430074, China [3]School of Mathematics and Physics, China University of Geosciences, Wuhan 430074, China. Correspondence to: Liangxiao Jiang <ljiang@cug.edu.cn>.

*Proceedings of the 43rd International Conference on Machine Learning*, Seoul, South Korea. PMLR 306, 2026. Copyright 2026 by the author(s).

instance-unaware and may repeatedly annotate similar instances. Model-centric strategies select instances based on model feedback, making them susceptible to cold starts. Although some studies (Yan et al., 2011; 2012) assume a semi-supervised setting to mitigate the impact of cold starts, they still require frequent model updates as new labels arrive, resulting in high computational costs. At present, no existing repeated annotation strategy can simultaneously avoid all of these limitations.

To address this, we propose MA³S, a novel model-agnostic active annotation strategy. First, to reduce label redundancy associated with offline procedure, MA³S estimates instance uncertainties with a margin-Beta distribution and updates these uncertainties online as new labels arrive. Second, to mitigate instance redundancy arising from instance-unaware designs, MA³S constructs a nearest-neighbor graph to propagate uncertainties across instances, thereby reducing repeated annotations of similar instances. Finally, to sidestep the drawbacks of the model-centric paradigm, MA³S actively selects instances based solely on the estimated uncertainties, without leveraging model feedback. In general, the contributions of this paper are as follows:

- We leverage a margin-Beta distribution to estimate and update instance uncertainties online, thereby enabling suitability for multi-class scenarios and reducing label redundancy associated with offline procedure.

- We construct a nearest-neighbor graph to propagate instance uncertainties, fully leveraging instance information in the attribute space and alleviating instance redundancy induced by instance-unaware designs.

- We propose MA³S, which actively selects the most valuable instances to annotate based on instance uncertainties, without relying on model feedback. This prevents data annotation from relying on the model.

- We conduct extensive experiments to validate the effectiveness of MA³S, and the results demonstrate that MA³S consistently and significantly outperforms existing repeated annotation strategies.

## 2. Related Work

In this section, we briefly review related work on repeated annotation strategies, organized into model-agnostic and model-centric categories, and discuss their limitations.

**Model-Agnostic Strategies.** The simplest repeated annotation strategy is generalized Round-robin (GRR) (Sheng et al., 2008), which relies solely on the count of crowd labels on each instance. In each round, GRR selects the instance with the fewest labels. As a simple and instance-unaware

strategy, it is commonly included as a baseline. Subsequently, Sheng et al. (2008) proposed an entropy-based strategy (ENTROPY) that calculates the entropy based on the collected labels of each instance, selecting the instance with the highest entropy in each round. Since entropy does not account for the number of labels and therefore cannot distinguish label sets $\{+, +, -\}$ and $\{+, +, +, +, -, -\}$, Sheng et al. later introduced label uncertainty-based strategy (LU). LU uses a standard Beta distribution to estimate label uncertainty for each instance, selecting the instance with the highest label uncertainty for annotation. Despite being online, ENTROPY and LU still remain instance-unaware and will inevitably annotate similar instances.

Coreset construction can be regarded as another class of model-agnostic strategies. Its core idea is to select a small yet representative subset (a coreset) from large-scale data, which can be used to approximate the solution on the full dataset (Jourdan et al., 2025; Feldman et al., 2025; 2016). Evidently, it entails instance selection and can therefore be adapted into a repeated annotation strategy. Considering the initial setup of crowdsourced data annotation, we primarily survey recent coreset construction methods for unsupervised tasks (Jiang et al., 2024; Huang et al., 2023). Using such methods, we can obtain a subset in a single pass and then annotate this subset with GRR. Clearly, the instance selection of coreset-based strategies cannot leverage information incorporated during annotation and is therefore offline.

**Model-Centric Strategies.** Active learning is a paradigm that dynamically queries labels for informative instances and incorporates them into the training set, enabling rapid and cost-effective performance gains through iterative model updates and progressive dataset expansion (Schmidt et al., 2025; Kou et al., 2025b; Cohn et al., 1994). This makes active learning naturally well aligned with crowdsourcing scenarios (He et al., 2022; Fang et al., 2014; 2012). Motivated by this connection, Sheng et al. (2008) proposed a model uncertainty-based strategy (MU) and a label and model uncertainty-based strategy (LMU). MU trains multiple models on the currently annotated instances, estimates each instance's model uncertainty via committee voting, and selects the instance with the highest uncertainty. LMU integrates LU and MU, jointly accounting for label uncertainty and model uncertainty.

We also review active learning from crowds (ALFC) methods, which also perform instance selection (Zhang et al., 2024; Zhong et al., 2015; Li et al., 2015; Yan et al., 2011). Most of these methods assume that worker abilities are accessible, which is difficult to satisfy in crowdsourcing scenarios. We adapt these methods to repeated annotation by removing their worker selection modules and retaining only the instance selection modules. Influenced by active learning, all of the above strategies are model-centric. Although

they avoid the limitations of being offline and instance-unaware, their performance is highly dependent on model performance and the quality of the initial annotations.

## 3. The Proposed Strategy

Based on the above review, it can be found that existing strategies cannot simultaneously avoid being offline, instance-unaware, and model-centric. To address this, we propose MA$^3$S, a model-agnostic active annotation strategy. Before presenting MA$^3$S in detail, we first provide a precise problem definition for crowdsourced repeated annotation.

### 3.1. Problem Definition

We consider a crowdsourced dataset $D^t = \{\boldsymbol{x}_i, \boldsymbol{L}_i^t\}_{i=1}^N$ that is currently being annotated. Here, $N$ denotes the number of instances, and $t$ denotes the annotation round. $\boldsymbol{x}_i$ is the $i$-th instance in $D^t$, which can be represented as $\{x_{im}\}_{m=1}^M$. $M$ is the dimension of attributes, and $x_{im}$ is the $m$-th attribute value of $\boldsymbol{x}_i$. $\boldsymbol{L}_i^t$ is the multiple noisy label set of $\boldsymbol{x}_i$ after $t$ rounds of annotation, which can be represented as $\{l_{ij}\}_{j=1}^t$. $l_{ij}$ is the label of $\boldsymbol{x}_i$ annotated in the $j$-th round. $l_{ij}$ takes a value from a fixed set $\{-1, c_1, \ldots, c_q, \ldots, c_Q\}$, where $Q$ is the number of classes, $c_q$ is the $q$-th class, and $-1$ means that $\boldsymbol{x}_i$ was not annotated in the $j$-th round. $y_i$ is the unknown true label of $\boldsymbol{x}_i$. When $l_{ij} \neq -1$, since crowd workers may assign an incorrect label, $l_{ij}$ and $y_i$ are not always the same. Considering that crowdsourced dataset is initially unannotated, $\boldsymbol{L}_i^t$ should be empty when $t = 0$. However, since existing model-centric strategies require a semi-supervised setup, we allow $\boldsymbol{L}_i^0 \neq \emptyset$ in this paper. Given $D^t$, a repeated annotation strategy is expected to identify the instance $\boldsymbol{x}^{t+1}$ with the highest benefit in the $(t+1)$-th round for annotation.

Besides, the above definitions are made under two assumptions. First, we assume that the annotation ability of each crowd worker is inaccessible. Since workers are recruited in open environments, it is unrealistic to know their annotation abilities in advance. Such a strict yet realistic assumption ensures that our MA$^3$S is applicable to any crowdsourcing scenario. Second, we assume that the unit cost of obtaining a label is fixed. This is a common assumption for data annotation (Hu & Zhang, 2018; Lin et al., 2016), ensuring that controlling annotation costs is equivalent to controlling the number of labels. Under this assumption, MA$^3$S can minimize the number of labels required to achieve high label accuracy, thereby reducing annotation costs.

### 3.2. Uncertainty Estimation

LU employs a standard Beta distribution to estimate instance uncertainties, successfully accounting for the effect of the number of labels on uncertainty, but it is applica-

ble only to binary scenarios. Our uncertainty estimation method is inspired by LU, extends LU to multi-class scenarios, and achieves stronger discriminative power. Specifically, after $t$ rounds of annotation, $\boldsymbol{L}_i^t$ can be converted into $\boldsymbol{\Phi} = \{N_1 + 1, \ldots, N_q + 1, \ldots, N_Q + 1\}$, where $N_q$ denotes the number of labels in $\boldsymbol{L}_i^t$ whose value is $c_q$. Since we assume that the annotation abilities of workers are inaccessible, the prior distribution over the true label is uniform in the interval $[0, 1]$. Consequently, the posterior probability over the true label follows $\boldsymbol{\Theta} \sim \text{Dirichlet}(\boldsymbol{\Phi})$. Here, $\boldsymbol{\Theta} = \{\theta_1, \ldots, \theta_q, \ldots, \theta_Q\}$ lies in the $Q - 1$ dimensional probability simplex, i.e., $\sum_{q=1}^Q \theta_q = 1$ and $\theta_q \geq 0$. Based on the probability density function (PDF) of the Dirichlet distribution, we can calculate $f(\boldsymbol{\Theta}; \boldsymbol{\Phi})$ as follows:

$$f(\boldsymbol{\Theta}; \boldsymbol{\Phi}) = \frac{\Gamma\left(\sum_{q=1}^Q (N_q + 1)\right)}{\prod_{q=1}^Q \Gamma(N_q + 1)} \prod_{q=1}^Q \theta_q^{N_q}, \quad (1)$$

where $\Gamma(N_q + 1)$ denotes the gamma function and its value equals $N_q!$ because $N_q + 1$ is a positive integer. Intuitively, we can compute the uncertainty of $\boldsymbol{L}_i^t$ as the tail probability below the decision threshold, i.e., the cumulative distribution function (CDF) of $\text{Dirichlet}(\boldsymbol{\Phi})$ at the decision threshold. Unfortunately, computing the CDF of a Dirichlet distribution requires integrating $f(\boldsymbol{\Theta}; \boldsymbol{\Phi})$ over the probability simplex, so there is generally no simple closed-form expression (Madsen et al., 2005; Ma et al., 2022).

For computational simplicity, we degenerate $\text{Dirichlet}(\boldsymbol{\Phi})$ to a Beta distribution because the Beta distribution admits a closed-form CDF (McDonald & Xu, 1995; Gupta & Nadarajah, 2004). However, the standard Beta distribution is not applicable to multi-class scenarios. Therefore, we need to design a more general version. Specifically, motivated by margin sampling (Joshi et al., 2009; Chen et al., 2022), we propose the margin-Beta distribution as follows:

$$\text{Beta}(N_m + 1, N_s + 1) = \frac{\Gamma(N_m + 1)\Gamma(N_s + 1)}{\Gamma(N_m + N_s + 2)}, \quad (2)$$

where $N_m + 1$ and $N_s + 1$ denote the maximum and the second maximum in $\boldsymbol{\Phi}$. In this way, the Beta distribution can handle both binary and multi-class scenarios. Ideally, workers would reach consensus on the unknown true class and the confusing class, so $N_m$ and $N_s$ are the most worth considering. In contrast, the other classes tend to aggregate more densely with random noisy labels. Under this premise, converting a multi-class scenario into multiple binary scenarios via one-vs-one or one-vs-rest and then applying the standard Beta distribution will increase the impact of noisy labels. Therefore, our proposed margin-Beta distribution is more practical. Once the margin-Beta distribution is defined, we can calculate its PDF as follows:

$$f(\theta; \alpha, \beta) = \frac{\theta^{\alpha-1}(1 - \theta)^{\beta-1}}{\text{Beta}(\alpha, \beta)}, \quad (3)$$

where $\alpha = N_m + 1$ and $\beta = N_s + 1$. Subsequently, we can calculate the CDF as follows:

$$I_\theta(\alpha, \beta) = \sum_{j=\alpha}^{\alpha+\beta-1} \binom{\alpha+\beta-1}{j} \theta^j (1-\theta)^{\alpha+\beta-1-j}. \quad (4)$$

For LU, because it is unclear whether $\alpha$ or $\beta$ is larger in the standard Beta distribution, the uncertainty is set to $min\{I_{0.5}(\alpha, \beta), 1 - I_{0.5}(\alpha, \beta)\}$. However, when the decision threshold equals 0.5, we have $I_{0.5}(\alpha, \alpha) = 0.5$, and thus LU cannot distinguish label sets $\{+, -\}$ and $\{+, +, +, -, -, -\}$. In contrast, for MA$^3$S, $\alpha = N_m + 1$ and $\beta = N_s + 1$, so $\alpha \geq \beta$. Therefore, we recommend choosing the decision threshold $\theta$ from the interval $(0, 0.5)$ and we set it to $0.4$ in this paper. When $\theta = 0.4$, because $\alpha \geq \beta$, $1 - I_{0.4}(\alpha, \beta)$ is greater than $I_{0.4}(\alpha, \beta)$. Therefore, the uncertainty of $\boldsymbol{L}_i^t$ is ultimately defined as follows:

$$U_i^t = I_{0.4}(\alpha, \beta). \quad (5)$$

According to the above definition of uncertainty, the following theorem can be proved to hold.

**Theorem 3.1.** *With $\alpha$ fixed, as $\beta$ increases, $U_i^t$ increases; with $\frac{\alpha}{\beta} = r$ fixed, as $\alpha$ increases, $U_i^t$ decreases.*

*Proof.* According to Eq. (4), we can obtain the following recurrence relation:

$$I_\theta(\alpha, \beta + 1) = I_\theta(\alpha, \beta) + \frac{\theta^\alpha (1-\theta)^\beta}{\beta \mathrm{Beta}(\alpha, \beta)}. \quad (6)$$

Here, since $\theta = 0.4$, $\alpha$ and $\beta$ are positive integers, we have $\frac{\theta^\alpha (1-\theta)^\beta}{\beta \mathrm{Beta}(\alpha, \beta)} > 0$. Therefore, $I_\theta(\alpha, \beta + 1) > I_\theta(\alpha, \beta)$. Hence, the first part of the theorem holds.

Next, fix $\frac{\alpha}{\beta} = r$, so that $\beta = \frac{\alpha}{r}$. The expectation and variance of the Beta distribution can be calculated as follows:

$$E_{\alpha, \beta} = \frac{\alpha}{\alpha + \beta} = \frac{r}{1+r}. \quad (7)$$

$$
\begin{aligned}
V_{\alpha, \beta} &= \frac{\alpha\beta}{(\alpha+\beta)^2(\alpha+\beta+1)} \\
&= \frac{1}{r(1+\frac{1}{r})^2(\alpha + \frac{\alpha}{r} + 1)}.
\end{aligned}
\quad (8)
$$

Therefore, it follows that $E_{\alpha, \beta}$ depends only on $r$, whereas $V_{\alpha, \beta}$ is inversely proportional to $\alpha$. Considering $\alpha_2 > \alpha_1$, then we can obtain $E_{\alpha_1, \beta_1} = E_{\alpha_2, \beta_2}$ and $V_{\alpha_1, \beta_1} > V_{\alpha_2, \beta_2}$. Moreover, since $\alpha = N_m + 1$ and $\beta = N_s + 1$, we can obtain $r \geq 1$. Therefore, $E_{\alpha, \beta} = \frac{r}{1+r} \geq 0.5$. Because MA$^3$S requires that $\theta$ belong to the interval $(0, 0.5)$, it follows that $E_{\alpha, \beta}$ belongs to the interval $(\theta, 1)$. Considering that a smaller variance makes the probability distribution more

tightly concentrated around the expectation, we can obtain the following relationship:

$$1 - I_\theta(\alpha_1, \beta_1) < 1 - I_\theta(\alpha_2, \beta_2). \quad (9)$$

By transforming Eq. (9), we can obtain $I_\theta(\alpha_1, \beta_1) > I_\theta(\alpha_2, \beta_2)$. Therefore, the second part of **Theorem 3.1** is proved. Furthermore, according to the second part of the theory, we can get $I_{0.4}(2, 2) > I_{0.4}(4, 4)$. As a result, $U_i^t$ can distinguish $\{+, -\}$ and $\{+, +, +, -, -, -\}$, and thus exhibits stronger discriminative power than LU.

$\square$

### 3.3. Instance Selection

Despite avoiding the offline procedure, $U_i^t$ shares LU's limitation of considering only $\boldsymbol{L}_i^t$ while ignoring $\boldsymbol{x}_i$, and thus remains instance-unaware. To address this, MA$^3$S constructs a nearest-neighbor graph on $D$ to propagate instance uncertainties. According to the manifold hypothesis (Narayanan & Mitter, 2010), data relationships are approximately linear within local neighborhoods. Therefore, MA$^3$S first calculates the Euclidean distance between $\boldsymbol{x}_i$ and $\boldsymbol{x}_j$ as follows:

$$d(\boldsymbol{x}_i, \boldsymbol{x}_j) = \sqrt{\sum_{m=1}^{M} (x_{im} - x_{jm})^2}. \quad (10)$$

Then, by sorting the distances between instances, MA$^3$S finds the $K$ nearest neighbors $\mathcal{N}_i$ for each instance $\boldsymbol{x}_i$. According to $\mathcal{N}_i$, MA$^3$S constructs a graph $\mathcal{A}$ as follows:

$$\mathcal{A}_{ij} = \begin{cases} 1, & \text{if } \boldsymbol{x}_i \in \mathcal{N}_j \text{ or } \boldsymbol{x}_j \in \mathcal{N}_i, \\ 0, & \text{otherwise.} \end{cases} \quad (11)$$

Next, MA$^3$S propagates instance uncertainties as follows:

$$\hat{U}_i^t = U_i^t + \frac{\sum_{j=1}^{N} \mathcal{A}_{ij} U_j^t}{\sum_{j=1}^{N} \mathcal{A}_{ij}}. \quad (12)$$

Finally, MA$^3$S actively selects the most valuable instance to annotate for the next round as follows:

$$\boldsymbol{x}^{t+1} = \arg\max_{\mathbf{x}_i \in D} \hat{U}_i^t. \quad (13)$$

In the $(t + 1)$-th annotation round, the multiple noisy label set of $\boldsymbol{x}^{t+1}$ is updated when the new label arrives, so MA$^3$S directly updates its uncertainty by Eq. (5). At the same time, to avoid an instance being repeatedly chosen because it has many neighbors, we reduce neighbor influence in Eq. (12) by averaging their uncertainties. For simplicity, in this paper MA$^3$S selects only one instance to annotate in each round. Notably, our MA$^3$S can be extended to batch annotation.

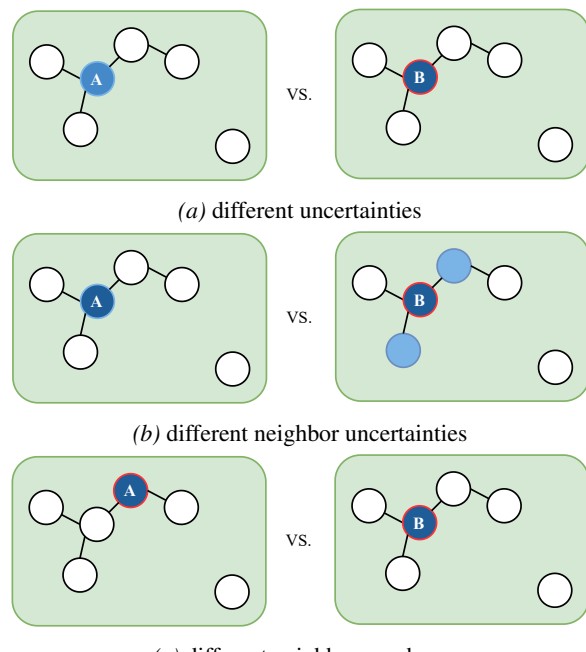

*(a) different uncertainties*

*(b) different neighbor uncertainties*

*(c) different neighbor numbers*

*Figure 1.* Case study for MA³S.

**Algorithm 1** The learning process of MA³S

**Input**: Crowdsourced dataset $D^0 = \{\boldsymbol{x}_i, \boldsymbol{L}_i^0\}_{i=1}^N$
**Parameter**: Annotation rounds $T$
**Output**: Annotated dataset $D^T = \{\boldsymbol{x}_i, \boldsymbol{L}_i^T\}_{i=1}^N$

1: **for** $i = 1$ to $N$ **do**
2:      Convert $\boldsymbol{L}_i^0$ into $\boldsymbol{\Phi}$ for instance $\boldsymbol{x}_i$.
3:      Initialize the uncertainty $U_i^0$ for $\boldsymbol{x}_i$ by Eq. (5).
4:      **for** $j = 1$ to $N$ **do**
5:          Calculate the distance $d(\boldsymbol{x}_i, \boldsymbol{x}_j)$ by Eq. (10).
6:      **end for**
7:      Sort the distances and find the neighbors $\mathcal{N}_i$ for $\boldsymbol{x}_i$.
8: **end for**
9: Construct the nearest-neighbor graph $\mathcal{A}$ by Eq. (11).
10: **for** $t = 1$ to $T$ **do**
11:      **for** $i = 1$ to $N$ **do**
12:          Propagate uncertainties to $\boldsymbol{x}_i$ by Eq. (12).
13:      **end for**
14:      Select the most valuable instance $\boldsymbol{x}^t$ by Eq. (13).
15:      Annotate $\boldsymbol{x}^t$ and update its $\boldsymbol{L}^t$ and $U^t$.
16: **end for**
17: **return** $D^T = \{\boldsymbol{x}_i, \boldsymbol{L}_i^T\}_{i=1}^N$

In the batch setting, we only need to modify Eq. (13) to select the top batch size instances with the highest uncertainties. After batch annotation, uncertainties are updated collectively using Eq. (5) and then propagated via Eq. (12). Due to the reduced number of propagation steps, MA³S with batch annotation will become more efficient. However, since uncertainty cannot be updated and propagated immediately after each individual annotation, its performance may be slightly weaker than in single-instance setting.

**Case Study.** To facilitate understanding of the above instance selection mechanism, we provide a case study in Figure 1. Here, instances are represented as nodes, and neighborhood relationships are represented as edges. We use node color intensity to represent an instance's uncertainty, and a red border to indicate MA³S's preferred selection. In case (a), Node A and Node B have identical neighborhoods, but Node A has lower uncertainty than Node B, so MA³S prefers annotating Node B. This shows that MA³S prioritizes higher-uncertainty instances, which benefits the online procedure by preventing repeated annotation of instances that already have sufficient labels. In case (b), Node A and Node B have identical uncertainties, but Node A has lower neighbor uncertainties than Node B. According to Eq. (12), Node A will have lower propagated uncertainty than Node B, so MA³S prefers annotating Node B. This shows that MA³S is instance-aware, preferring instances with more uncertain neighborhoods. In addition, once a node's uncertainty decreases, the propagated uncertainties of its neighbors also decrease, which lowers their chances of

being selected and thus effectively limits MA³S from annotating redundant instances. In case (c), Node A and Node B have identical uncertainties and neighbor uncertainties, but Node A has fewer neighbors than Node B. Since the neighbors' uncertainties are averaged during propagation, the two nodes have the same chance of being selected by MA³S. This prevents MA³S from focusing solely on instances with many neighbors. The above instance selection mechanism ensures that MA³S can choose the most valuable instances for annotation.

**Time Complexity Analysis.** The whole process of MA³S is shown in Algorithm 1. In Algorithm 1, line 2 converts $\boldsymbol{L}_i^0$ into $\boldsymbol{\Phi}$ for instance $\boldsymbol{x}_i$ and its time complexity is $O(Q)$. Line 3 initializes the uncertainty $U_i^0$ for $\boldsymbol{x}_i$ and its time complexity is $O(Q \log Q)$. Lines 4-6 calculate the distances and their time complexity is $O(NM)$. Line 7 finds the neighbors $\mathcal{N}_i$ for $\boldsymbol{x}_i$ and its time complexity is $O(N \log N)$. Since $N$ is much greater than $Q$ and $M$, the time complexity of lines 1-8 is $O(N^2 \log N)$. Line 9 constructs the nearest-neighbor graph $\mathcal{A}$ and its time complexity is $O(N^2 K)$. Lines 11-13 propagate uncertainties and their time complexity is $O(N^2)$. Line 14 selects the most valuable instance $\boldsymbol{x}^t$ and its time complexity is $O(N)$. Line 15 updates $U^t$ for $\boldsymbol{x}^t$ and its time complexity is $O(Q \log QT^2)$. In practice, it is unrealistic for all labels to be annotated to the same instance. Each instance typically receives few labels, so $O(Q \log QT^2)$ is negligible. Considering only the highest-order terms, the overall time complexity of MA³S is $O(N^2 \log N + N^2 T)$.

*Table 1.* Statistics of four real-world datasets.

| Dataset | *Income* | *Leaves* | *Music* | *LabelMe* |
|---|---|---|---|---|
| #Instances | 600 | 384 | 700 | 1000 |
| #Labels | 6000 | 3840 | 2946 | 2547 |
| #Attributes | 10 (nominal) | 64 (numeric) | 31 (numeric) | 512 (numeric) |
| #Classes | 2 | 6 | 10 | 8 |

# 4. Experiments and Results

In this section, we conduct extensive experiments to validate the effectiveness of MA$^3$S. First, we report the experimental setup. Then, we validate the properties and potential advantages of MA$^3$S using a synthetic dataset. Finally, we comprehensively evaluate the effectiveness and significance of MA$^3$S on multiple real-world datasets.

## 4.1. Experimental Setup

As a repeated annotation strategy, the key baselines for MA$^3$S include GRR, ENTROPY, LU, MU, and LMU, which are discussed collectively in Sheng et al. (2008). In addition, we implement a classical ALFC method (Fang et al., 2012) and a recent DPP-based (determinantal point process) coreset construction method (Bardenet et al., 2024) as annotation strategies for comparison. Among these baselines, MU, LMU, and ALFC are all model-centric strategies. For LU and LMU, we compare them only in binary scenarios. For ALFC, we remove its worker selection module while retaining its instance selection module. For all model-centric strategies, we use the same random forest (Breiman, 2001) with 10 trees. For DPP, we set the coreset size to $\frac{N}{10}$ and annotate the coreset using GRR. All strategies are implemented in Python, and their core settings follow the corresponding papers. For MA$^3$S, the default number of neighbors $K$ is set to 3.

To analyze the properties and effectiveness of MA$^3$S, we construct both synthetic and real-world experiments. In the synthetic setting, for ease of visualization, we first create a dataset with two attributes and two classes (90 instances per class). Then, we control instance uncertainty via two factors: the total number of labels and the proportion of incorrect labels. Specifically, we divide the synthetic instances into six categories and assign diverse initial multiple noisy label sets as follows: $\mathcal{I}_1$: {+}, $\mathcal{I}_2$: {+, −}, $\mathcal{I}_3$: {+, +, −}, $\mathcal{I}_4$: {+, +, −, −}, $\mathcal{I}_5$: {+, +, +, +, −}, $\mathcal{I}_6$: {+, +, +, +, −, −}. Here, "+" denotes the true label of an instance, and "−" denotes a randomly assigned incorrect label. The instances are sequentially assigned to these categories, with 30 instances per category. Among these categories, ($\mathcal{I}_2$, $\mathcal{I}_4$) and ($\mathcal{I}_3$, $\mathcal{I}_6$) form controlled pairs: they share the same proportion of incorrect labels but differ in the number of labels. Finally, during each annotation round, we sample a worker quality $p$ uniformly from [0.55, 0.75]. The selected instance receives a correct label with probability $p$. Otherwise, it receives a

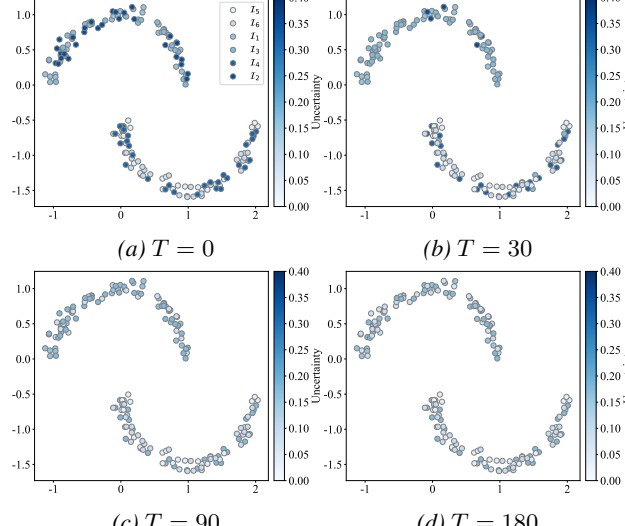

*(a)* $T = 0$      *(b)* $T = 30$

*(c)* $T = 90$      *(d)* $T = 180$

*Figure 2.* Evolution of instance uncertainties on the synthetic dataset as annotation rounds increase.

randomly incorrect label. Using this setup, we analyze the properties and potential advantage of MA$^3$S.

In the real-world setting, we compare MA$^3$S against all baselines on four widely used crowdsourced datasets: *Income*, *Leaves*, *Music*, and *LabelMe* (Zhang et al., 2025b; Wu et al., 2025). These datasets were collected on Amazon Mechanical Turk (AMT), and their statistics are reported in Table 1. Here, "#Instances" denotes the number of instances. Considering that Eq. (10) supports only numerical attributes, we preprocess datasets with nominal attributes using LabelEncoder before experiments. Unlike the synthetic setting, we directly use each instance's current multiple noisy label set as its initial label set. The annotation procedure follows the same protocol as in the synthetic experiments. After annotation by different strategies, we apply majority voting (MV) (Sheng et al., 2008) to infer each instance's unknown true label from its multiple noisy label set and report the label accuracy. Here, label accuracy is defined as the proportion of instances whose inferred labels match their unknown true labels. To mitigate the impact of randomness, we independently repeat the experiments on each real-world dataset ten times and report the averaged results.

## 4.2. Synthetic Experiments

**Efficiency.** In synthetic experiments, we first evaluate the annotation efficiency of MA$^3$S. As shown in Figure 2, we examine how instance uncertainties in the synthetic dataset evolve with the annotation rounds $T$. In Figure 2, colors encode uncertainty (darker = higher uncertainty). When $T = 0$, Figure 2a shows the original synthetic dataset, and we also visualize the category of instances for each uncertainty level. We find that MA$^3$S can distinguish ($\mathcal{I}_2$,

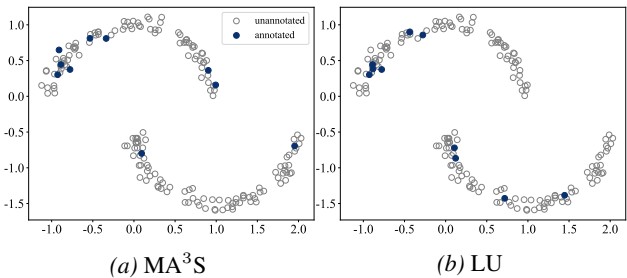

*(a)* MA$^3$S          *(b)* LU

*Figure 3.* Instance selection results with $T = 10$.

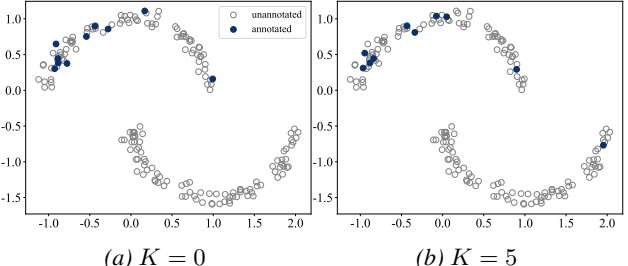

*(a)* $K = 0$          *(b)* $K = 5$

*Figure 5.* Instance selection results of MA$^3$S across different $K$.

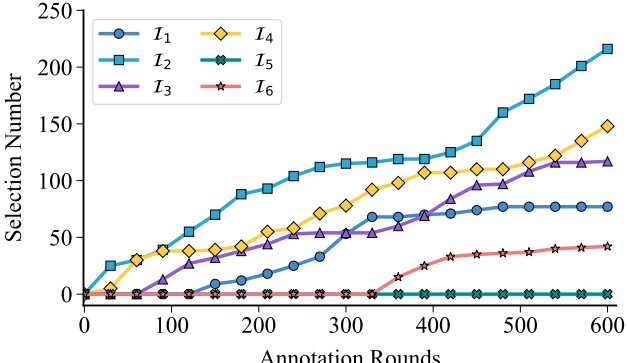

*Figure 4.* Selection number of MA$^3$S in each category across annotation rounds.

$\mathcal{I}_4$). MA$^3$S considers $\mathcal{I}_2$ to be more uncertain than $\mathcal{I}_4$ because it receives fewer labels, which aligns better with reality. It can also be seen from Figure 2 that the most uncertain instances have their uncertainties reduced rapidly by $T = 30$. Moreover, high-uncertainty instances are almost absent by $T = 90$, and the overall uncertainty of the dataset is markedly lower by $T = 180$. These results indicate that MA$^3$S requires only about $N$ annotations to quickly drive down uncertainty on the synthetic dataset. For comparison, we also visualize the instance uncertainties of the synthetic dataset at $T = 90$ after annotation with other baselines. Due to the limited pages, these results are provided in **Appendix A**. It can be seen that, under the same annotation rounds, MA$^3$S achieves a more pronounced reduction in the overall uncertainty of the synthetic dataset than other baselines, demonstrating its superior annotation efficiency.

**Awareness.** The previous experiments demonstrate the annotation efficiency of MA$^3$S, but they do not capture its instance awareness and therefore cannot fully reveal the fundamental differences between MA$^3$S and other strategies. To showcase the potential advantages of instance awareness, we further examine the instance selection behavior of MA$^3$S. Figure 3 shows the ten instances annotated by MA$^3$S when $T = 10$. For comparison, we also present the ten instances selected by the instance-unaware LU. As shown in Figures 2a and 3, MA$^3$S selects more instances from cat-

egory $\mathcal{I}_2$ and fewer from $\mathcal{I}_4$, whereas LU selects roughly equal numbers from both categories. This highlights the advantage of MA$^3$S in distinguishing between $\mathcal{I}_2$ and $\mathcal{I}_4$. Moreover, MA$^3$S selects some instances from $\mathcal{I}_4$ because it jointly considers the uncertainties of each instance and its neighbors, thereby choosing the most valuable instances in each round and achieving higher annotation gains. In contrast, instance-unaware strategies may annotate similar instances within the same neighborhood. To more thoroughly demonstrate the awareness of MA$^3$S, Figure 4 visualizes the number of instances it selects from each category across annotation rounds. We observe that, as the rounds progress, MA$^3$S generally selects instances in order of increasing uncertainty. Since the uncertainty of an instance changes after receiving new labels, MA$^3$S also has opportunities to select lower-uncertainty instances in later iterations. **Appendix B** presents the corresponding results for the other baselines. We can observe that none of them can distinguish instances across categories as effectively as MA$^3$S.

**Sensitivity.** The efficiency and awareness of MA$^3$S primarily rely on uncertainty estimation and uncertainty propagation. For uncertainty estimation, we establish the properties of $U_i^t$ in **Theorem 3.1**. When the parameter $\theta \in (0, 0.5)$, **Theorem 3.1** holds, indicating that MA$^3$S is insensitive to $\theta$. For uncertainty propagation, the neighborhood size is controlled by the parameter $K$. To assess the sensitivity of MA$^3$S to $K$, we conduct a parameter sensitivity analysis. We fix $T$ at 10 and examine the instance selection behavior of MA$^3$S when $K$ is set to 0 and 5, respectively. The results are shown in Figure 5. We can find that when $K = 0$, MA$^3$S is no longer instance-aware. It annotates only instances from $\mathcal{I}_2$. Moreover, without considering neighbor uncertainties, MA$^3$S may annotate similar instances. For other values of $K$, the behavior of MA$^3$S remains largely unchanged, since Eq. (12) averages the influence of neighbors. However, to preserve local awareness, we recommend setting $K$ to a small value. By default, we set $K = 3$ in this paper.

### 4.3. Real-World Experiments

**Effectiveness.** Although the synthetic experiments demonstrate some properties and potential advantages of MA$^3$S,

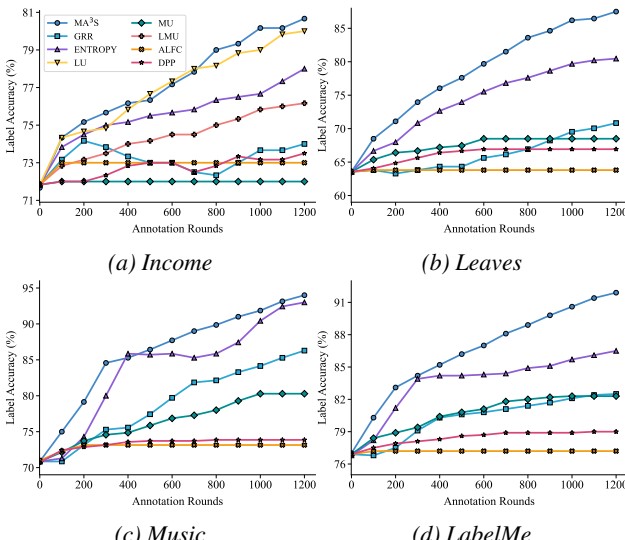

*Figure 6.* Label accuracies (%) of each dataset after applying each annotation strategy across different annotation rounds.

we have not yet conducted a comprehensive comparison between MA$^3$S and other baselines to establish its effectiveness and significance. Moreover, to facilitate the control of instance uncertainty, we construct the synthetic experiments only for the binary scenarios. Therefore, we further conduct a comprehensive comparison of MA$^3$S with other annotation strategies on multiple real-world datasets. In these datasets, the initial multiple noisy label set of each instance is obtained from crowd workers rather than manual simulation, better reflecting the actual annotating conditions. Meanwhile, these dataset are collected from both binary and multi-class scenarios, enabling a comprehensive validation of MA$^3$S's effectiveness. Among them, since LU and LMU can only handle binary scenarios, they have experimental results only on the *Income* dataset. Figure 6 shows the label accuracy for each dataset after applying each annotation strategy across different annotation rounds. Based on these results, we can draw the following highlights:

- MA$^3$S achieves the most competitive label accuracies across all datasets and annotation rounds, validating the effectiveness of MA$^3$S in data annotation by simultaneously addressing the three types of limitations.

- Both GRR, ENTROPY and LU are instance-unaware strategies, and they achieve suboptimal label accuracies, indicating that they cannot identify the most valuable instances to annotate in each round.

- Both MU and ALFC are model-centric, and their performance converges prematurely. This reflects that they are highly dependent on the model's convergence and the quality of the initial annotated data.

*Table 2.* Wilcoxon test results on the *Income* dataset.

| Dataset | GRR | ENTROPY | LU | MU | LMU | ALFC | DPP | MA$^3$S |
|---|---|---|---|---|---|---|---|---|
| GRR | - | ○ | ○ | ● | ○ | ● | ● | ○ |
| ENTROPY | ● | - | ○ | ● | ● | ● | ● | ○ |
| LU | ● | ● | - | ● | ● | ● | ● | ○ |
| MU | ○ | ○ | ○ | - | ○ | ○ | ○ | ○ |
| LMU | ● | ○ | ○ | ● | - | ● | ● | ○ |
| ALFC | ○ | ○ | ○ | ● | ○ | - | ○ | ○ |
| DPP | ○ | ○ | ○ | ● | ○ | ● | - | ○ |
| MA$^3$S | ● | ● | ● | ● | ● | ● | ● | - |

*Table 3.* Wilcoxon test results on the *Leaves* dataset.

| Dataset | GRR | ENTROPY | MU | ALFC | DPP | MA$^3$S |
|---|---|---|---|---|---|---|
| GRR | - | ○ | ○ | ● | | ○ |
| ENTROPY | ● | - | ● | ● | ● | ○ |
| MU | ● | ○ | - | ● | ● | ○ |
| ALFC | ○ | ○ | ○ | - | ○ | ○ |
| DPP | | ○ | ○ | ● | - | ○ |
| MA$^3$S | ● | ● | ● | ● | ● | - |

*Table 4.* Wilcoxon test results on the *Music* dataset.

| Dataset | GRR | ENTROPY | MU | ALFC | DPP | MA$^3$S |
|---|---|---|---|---|---|---|
| GRR | - | ○ | ● | ● | ● | ○ |
| ENTROPY | ● | - | ● | ● | ● | ○ |
| MU | ○ | ○ | - | ● | ● | ○ |
| ALFC | ○ | ○ | ○ | - | ○ | ○ |
| DPP | ○ | ○ | ○ | ● | - | ○ |
| MA$^3$S | ● | ● | ● | ● | ● | - |

- Although the results of DPP are generally better than those of model-centric strategies, it is offline and can annotate only a portion of the dataset, its improvement to the whole dataset is still limited.

**Significance.** Figure 6 reports only the average results over 10 repeated runs. Because we use different annotation rounds on each dataset, and each round is repeated 10 times, we obtain more than 100 experimental results per dataset. To further determine whether MA$^3$S's effectiveness is statistically significant, we apply the Wilcoxon signed-rank test (Demsar, 2006; Jansen et al., 2023) to compare each pair of strategies on each dataset. Tables 2-5 summarize the Wilcoxon test results on each dataset. In these tables, the symbol ● indicates that the strategy in the corresponding row significantly outperforms the strategy in the corresponding column, while ○ indicates the opposite. The significance levels for the lower and upper diagonals are 0.05 and 0.1, respectively. Based on these results, we can find that MA$^3$S consistently and significantly outperforms all baselines in both binary and multi-class scenarios, which strongly supports the effectiveness and superiority of MA$^3$S.

**Reasonableness.** MA$^3$S mainly improves repeated annotation performance through uncertainty estimation and uncertainty propagation. To further verify the individual

*Table 5.* Wilcoxon test results on the *LabelMe* dataset.

| Dataset | GRR | ENTROPY | MU | ALFC | DPP | MA³S |
|---|---|---|---|---|---|---|
| GRR | - | ○ | ○ | ● | ● | ○ |
| ENTROPY | ● | - | ● | ● | ● | ○ |
| MU | ● | ○ | - | ● | ● | ○ |
| ALFC | ○ | ○ | ○ | - | ○ | ○ |
| DPP | ○ | ○ | ○ | ● | - | ○ |
| MA³S | ● | ● | ● | ● | ● | - |

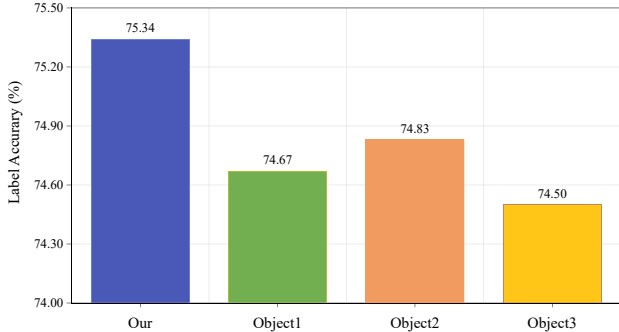

*Figure 7.* Label accuracy (%) achieved by MA³S and its variants on the *Income* dataset.

contributions of these two key components, we construct a set of ablation studies. Specifically, we design three ablated variants for MA³S (Our), denoted as Object1, Object2, and Object3. Here, Object1 degrades the uncertainty estimation to be the same as LU, Object2 removes uncertainty propagation, and Object3 combines both modifications (i.e., equivalent to LU). Considering that LU cannot handle multiclass scenarios, we conduct the ablation studies only on the *Income* dataset. We evaluate the label accuracy (%) of the *Income* dataset after applying each variant with 200 annotation rounds, and the results are shown in Figure 7. According to Figure 7, it can be observed that removing any component degrades the performance of MA³S. When both components are removed, the worst label accuracy is obtained. Therefore, we can validate the contribution of each key component in MA³S and these results demonstrate the reasonableness of MA³S.

## 5. Conclusion

In this paper, we propose a model-agnostic active repeated annotation strategy called MA³S. MA³S introduces a margin-Beta distribution to effectively estimate instance uncertainty, which allows it to actively and online select instances for annotation. Subsequently, MA³S achieves instance awareness by propagating uncertainties over a nearest-neighbor graph, which further enhances the annotation efficiency. Ultimately, MA³S effectively overcomes the limitations of existing annotation strategies, and both extensive theoretical analysis and experimental results validate

its effectiveness and significance.

However, although instance awareness is achieved by propagating neighbor uncertainties during instance selection, the new labels received by selected instances are not shared with their neighbors in current MA³S. In addition, propagating uncertainty via the nearest-neighbor graph incurs an $O(N^2)$ time complexity. While acceptable, this leaves room for further optimization. In future work, we will improve MA³S along these two directions.

## Acknowledgments

The work was partially supported by the National Natural Science Foundation of China (62276241), the Open Project Program of Yunnan Key Laboratory of Intelligent Systems and Computing (ISC25Z01) and the Scientific and Technological Development Scheme of Jilin Province, China (20250205059GH).

## Code and Data Availability Statement

The source code of MA³S and the datasets used in this paper are available from the following websites: https://github.com/jiangliangxiao/MA³S.

## Impact Statement

This paper presents work whose goal is to advance the field of Machine Learning. There are many potential societal consequences of our work, none which we feel must be specifically highlighted here.

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

## A. Overall uncertainty of the synthetic dataset after annotation with other baselines.

*Figure 8.* Overall uncertainty of the synthetic dataset at $T = 90$ after annotation with other baselines.

Similar to Figure 2c, we visualize the overall uncertainty of the synthetic dataset after annotation by the other baselines under the same conditions. From Figure 8, we can observe that GRR, by always selecting the instances with the least labels, can rapidly reduce the uncertainties of category $\mathcal{I}_1$. However, it does not distinguish among instances with the same number of labels, so some highly uncertain instances remain in Figure 8a. Both ENTROPY and LU estimate the uncertainties of the label sets and achieve suboptimal performance. MU, LMU, and ALFC are all model-centric strategies, and they perform poorly, reflecting their strong dependence on the model itself. When the model underperforms, updates per round are marginal, or the initial annotated data are of low quality, model-centric strategies tend to degrade. DPP is a coreset-based strategy. Because it can only annotate a subset, it yields the least reduction in overall uncertainty. These results further support the superior annotation efficiency of MA³S.

## B. Selection number of baselines in each category across annotation rounds.

Similar to Figure 4, we visualize the number of instances other baselines select from each category across annotation rounds. According to Figure 9, we can examine the ability of each baseline to distinguish every category in the synthetic dataset. First, although GRR can distinguish categories, its selection order is $\mathcal{I}_1 \rightarrow \mathcal{I}_2 \rightarrow \mathcal{I}_3 \rightarrow \mathcal{I}_4 \rightarrow \mathcal{I}_5 \rightarrow \mathcal{I}_6$, i.e., by the number of labels received per instance. However, according to Figure 2a, the uncertainty order should be

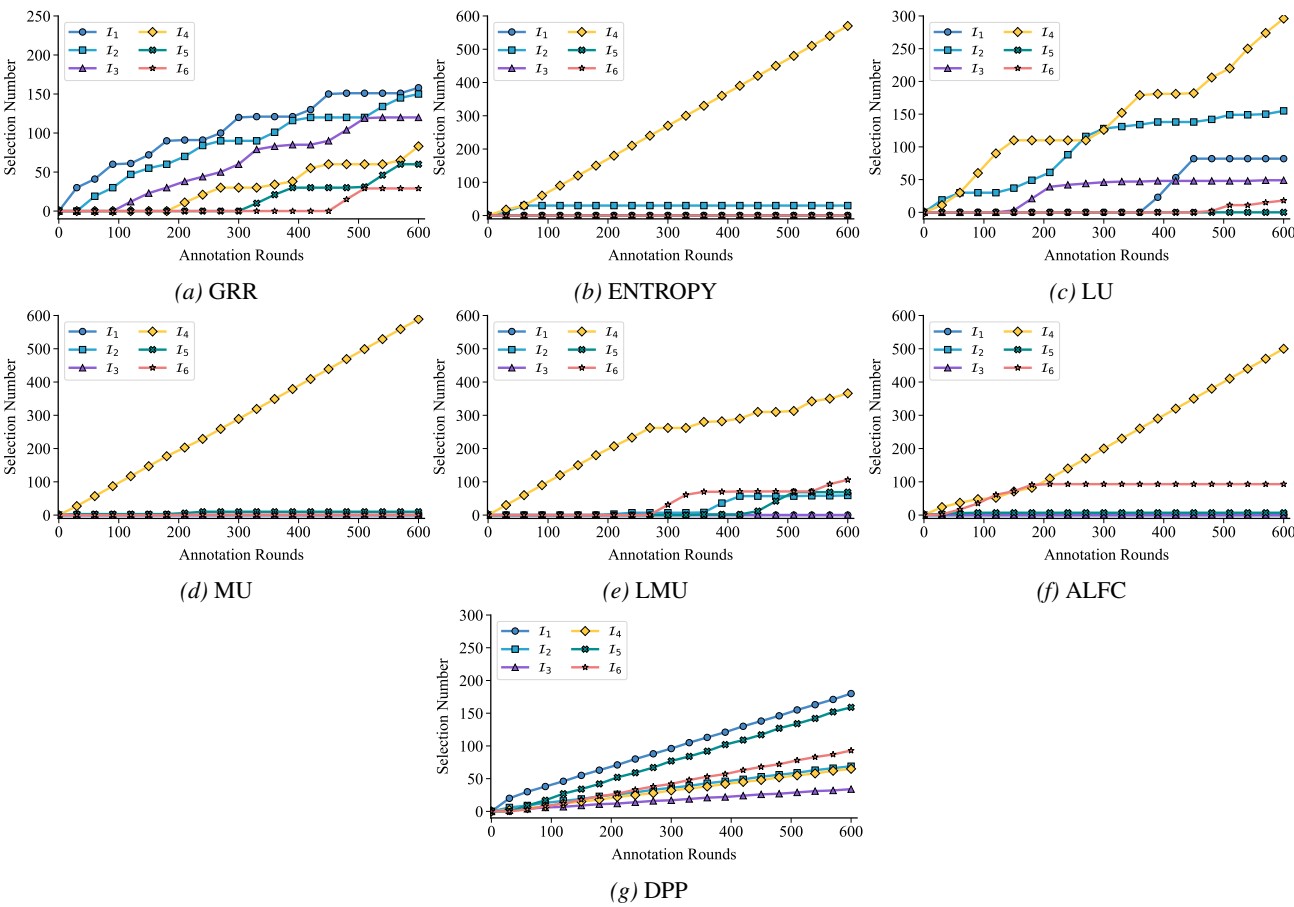

*Figure 9.* Selection number of baselines in each category across annotation rounds.

$\mathcal{I}_2 \rightarrow \mathcal{I}_4 \rightarrow \mathcal{I}_3 \rightarrow \mathcal{I}_1 \rightarrow \mathcal{I}_6 \rightarrow \mathcal{I}_5$. Therefore, GRR does not truly distinguish instances across categories. Second, ENTROPY cannot distinguish $\mathcal{I}_2$ from $\mathcal{I}_4$. At early rounds, it selects instances from $\mathcal{I}_2$ and $\mathcal{I}_4$ indistinguishably. At later rounds, it favors $\mathcal{I}_4$. This occurs because instances in $\mathcal{I}_4$ have more labels, so their entropy decreases more slowly after receiving new labels. Consequently, ENTROPY requires more rounds before it begins to attend to other categories. Third, LU is the second-best strategy. Aside from its inability to distinguish $\mathcal{I}_2$ from $\mathcal{I}_4$, it effectively separates the remaining categories in order of uncertainty. This highlights the advantage of $U_i^t$ in MA$^3$S over LU and empirically reaffirms **Theorem 3.1**. Fourth, model-centric strategies all suffer from the same problem: they overweight a single category (in effect, a single instance), most notably MU. After the model reaches convergence, model-centric strategies repeatedly select the same instance. On this instance, the converged model remains persistently uncertain, and subsequent updates have little effect. This reflects the drawback that model-centric strategies are highly dependent on the model and the quality of initial annotations. Finally, because DPP annotates only a representative subset, it repeatedly selects instances from this subset. As a result, the number of annotated instances in each category increases roughly proportionally with the number of rounds. Consequently, DPP also fails to distinguish instances across categories. In summary, none of the existing strategies distinguish instances with different levels of uncertainty as effectively as MA$^3$S, further indicating its practical potential.

