# OpenReview forum: "MA$^3$S: Model-Agnostic Active Annotation Strategy for Crowdsourcing"
_ICML.cc/2026/Conference — ICML 2026 regular_

### Official Review · Reviewer_q5sj · 2026-02-27

**Soundness:** 4
**Presentation:** 3
**Significance:** 4
**Originality:** 3
**Overall Recommendation:** 5
**Confidence:** 4

**Summary:**

In crowdsourcing scenarios, repeated annotation can introduce instance- or label-level redundancy. Existing repeated annotation strategies confront from three limitations: the offline procedure, instance-unaware design, and the model-centric paradigm. To addresses these limitations, this paper proposes a model-agnostic active annotation strategy, simply denoted by MA$^3$S. The extensive experiments on a synthetic dataset and four real-world crowdsourced datasets show that MA$^3$S significantly outperforms existing annotation strategies used to compare.

**Compliance With Llm Reviewing Policy:**

Affirmed.

**Final Justification:**

All questions resolved. I'm keeping my positive score.

**Key Questions For Authors:**

The same as the weaknesses above.

**Limitations:**

Yes

**Strengths And Weaknesses:**

Strengths:

1. To simultaneously address three limitations confronting existing repeated annotation strategies, this paper proposes MA$^3$S, a novel model-agnostic active annotation strategy.

2. The paper addresses an interesting and important problem in crowdsourcing learning. And the paper is well organized and written, with clearly presented analysis.

3. The experiments are extensive and convinced. The experimental results on a synthetic dataset and four real-world crowdsourced datasets show that MA$^3$S significantly outperforms existing annotation strategies used to compare.

Weaknesses:

1. “Model-agnostic” is highlighted as a main contribution in this paper, but its definition needs further clarification. The proposed strategy still relies on distance metrics and graph-based propagation, which are also modeling assumptions. It is recommended to clarify that “model-agnostic” refers specifically to “not relying on predictive model feedback”.

2. Although the main text states that Theorem 1 holds when $\theta \in (0,0.5) $, the direct choice of $\theta = 0.4$ in Equation (5) should be supported by more experimental evidence. It is recommended to add a sensitivity analysis of $\theta$ in the experimental section.

3. There are few typos that need correction. For example, in Equation (10), $(x_{im}-x^2_{jm})$ should be $(x_{im}-x_{jm})^2$. Please carefully proofread the entire paper and improve the writing.

---

> ### Author Rebuttal · Authors · 2026-03-30
>
> **W1:** “Model-agnostic” is highlighted as a main contribution in this paper, but its definition needs further clarification. The proposed strategy still relies on distance metrics and graph-based propagation, which are also modeling assumptions. It is recommended to clarify that “model-agnostic” refers specifically to “not relying on predictive model feedback”.
>
> **Author Response to W1:** Thanks for your valuable comments. In this paper, *“model-agnostic”* mainly means that MA$^3$S does not rely on model feedback for instance selection. In the revised version, we will adopt the reviewer’s suggestion and clarify that *“model-agnostic”* specifically refers to *“not relying on predictive model feedback.”* Thanks again for your valuable comments.
>
> **W2:** Although the main text states that Theorem 1 holds when $\theta \in (0,0.5)$, the direct choice of $\theta = 0.4$ in Equation (5) should be supported by more experimental evidence. It is recommended to add a sensitivity analysis of $\theta$ in the experimental section.
>
> **Author Response to W2:** Thanks for your valuable comments. The theoretical analysis in this paper indicates that any value of $\theta$ within the range (0, 0.5) is valid. Therefore, we simply set it to 0.4. To address the reviewer’s concern and provide experimental evidence for this choice, we conduct a sensitivity analysis to examine the impact of different $\theta$ values on MA$^3$S. Specifically, we vary $\theta$ from 0.05 to 0.45 with a step of 0.05 and evaluate the label accuracies (\%) of four real-world datasets after applying MA$^3$S with 200 annotation rounds, and the results are as follows:
> |$\theta$|0.05|0.10|0.15|0.20|0.25|0.30|0.35|0.40|0.45|
> |--|--|--|--|--|--|--|--|--|--|
> |*Income*|**75.83**|74.67|75.33|75.16|74.83|74.50|74.83|75.34|75.67|
> |*Leaves*|71.35|71.61|72.40|71.61|71.88|**72.92**|71.61|72.14|71.35|
> |*Music*|77.86|78.00|77.57|78.71|78.43|78.71|77.57|78.00|**78.86**|
> |*LabelMe*|82.30|83.10|82.30|**83.30**|82.90|82.20|82.90|82.70|82.30|
> ||
>
> Based on these results, we can observe that MA$^3$S is not sensitive to variations in $\theta$. In the revised version, we will include the above sensitivity analysis for $\theta$. Thanks again for your valuable comments.
>
> **W3:** There are few typos that need correction. For example, in Equation (10), $(x_{im}-x_{jm}^2)$ should be $(x_{im}-x_{jm})^2$. Please carefully proofread the entire paper and improve the writing.
>
> **Author Response to W3:** Thanks for your valuable comments. In Equation (10), the square should be placed outside the parentheses. We will update Equation (10) as follows:
> $$
> d({x}\_i,{x}\_j) = \sqrt{\sum\_{m=1}^{M}(x\_{im}-x\_{jm})^2}.
> $$
>
> Additionally, we will double-check and improve the writing of our paper. Thanks again for your valuable comments.

---

> > ### Author Rebuttal · Reviewer_q5sj · 2026-04-02
> >
> > All questions resolved. I'm keeping my positive score.

---

### Official Review · Reviewer_SBcy · 2026-02-27

**Soundness:** 3
**Presentation:** 3
**Significance:** 3
**Originality:** 3
**Overall Recommendation:** 5
**Confidence:** 4

**Summary:**

This paper proposes a model-agnostic active repeated annotation strategy for crowdsourcing that aims to reduce both label redundancy (over-annotating easy instances) and instance redundancy (re-annotating similar instances). The proposed strategy first estimates per-instance uncertainty online using a “margin-Beta” reduction of the Dirichlet posterior (taking the largest and second-largest class counts to form a Beta (α, β) and using its CDF at a fixed threshold). Then, it propagates uncertainties over a KNN graph to incorporate instance similarity and avoid redundant selections. Finally, it selects instances purely based on these propagated uncertainties, without relying on model training or model feedback. Experiments on a synthetic dataset and four real-world AMT datasets suggest the new strategy can achieve higher label accuracy (via majority vote) with fewer total annotations.

**Compliance With Llm Reviewing Policy:**

Affirmed.

**Final Justification:**

My concerns have been adequately addressed. I maintain the positive score.

**Key Questions For Authors:**

See the weaknesses above.

**Limitations:**

yes

**Strengths And Weaknesses:**

Strengths:
1. The paper explicitly targets three practical shortcomings in repeated annotation, including offline selection, instance unawareness, and model-centric dependence, and then proposes a new strategy. The new strategy does not require training or updating a predictive model, avoiding cold-start sensitivity and frequent retraining costs.
2. The new strategy reduces the Dirichlet distribution to a two-parameter Beta distribution via top 2 class counts (margin), which is a simple, computationally light alternative to intractable Dirichlet CDFs. The KNN-based uncertainty propagation encourages diversity in selected instances and limits local redundancies.
3. The experiments evaluate the new strategy against classical repeated annotation baselines (GRR/ENTROPY/LU), model-centric strategies (MU/LMU/ALFC), and a coreset method (DPP) on both synthetic and multiple real-world datasets. The results demonstrate that the new strategy significantly outperforms these benchmarks.

Weaknesses:
1. For computational simplicity, the paper reduces the Dirichlet distribution to a Beta distribution, which is reasonable. However, this Beta distribution should not be called “general Beta distribution”, which is potentially confusing. I recommend using a more precise term, such as “margin-Beta distribution”.
2. In instance selection section, uncertainty propagation is based only on a KNN graph, where neighbor weights are limited to 0 or 1. It is necessary to discuss and compare more sophisticated graph construction methods or neighbor weighting methods.
3. The paper currently discusses only single-instance annotations, i.e., selecting one instance per round for labeling, which can be limited in real-world applications. It is recommended to discuss the results of the new strategy under batch annotation, including how the strategy is adapted and how it performs when selecting multiple instances per round.

---

> ### Author Rebuttal · Authors · 2026-03-30
>
> **W1:** For computational simplicity, the paper reduces the Dirichlet distribution to a Beta distribution, which is reasonable. However, this Beta distribution should not be called “general Beta distribution”, which is potentially confusing. I recommend using a more precise term, such as “margin-Beta distribution”.
>
> **Author Response to W1:** Thanks for your valuable comments. Considering that the traditional Beta distribution is limited to binary scenarios, while our Equation (2) can handle both binary and multi-class scenarios, we originally used *"general Beta distribution"* for distinction. As the reviewer suggested, *“margin-Beta distribution”* is indeed a more precise representation. In the revised version, we will adopt this suggestion and use *“margin-Beta distribution”* to denote our contribution. Thanks again for your valuable comments.
>
> **W2:** In instance selection section, uncertainty propagation is based only on a KNN graph, where neighbor weights are limited to 0 or 1. It is necessary to discuss and compare more sophisticated graph construction methods or neighbor weighting methods.
>
> **Author Response to W2:** Thanks for your valuable comments. Indeed, as the reviewer pointed out, more sophisticated graph construction or neighbor weighting methods may be more beneficial for uncertainty propagation, but they typically come with higher computational complexity. For simplicity, the current manuscript adopts a KNN graph with 0/1 weights. To further address the reviewer’s concern, we construct a set of comparative experiments to evaluate the impact of neighbor weighting on MA$^3$S. Specifically, we first transform the Euclidean distance into a weight using the following Gaussian kernel (with $\sigma$ set to 0.5):
> $$
> w_{ij} = exp(\frac{-d(x_i, x_j)^2} { 2σ^2} ).
> $$
> Then, we refine the uncertainty propagation function of MA$^3$S as follows:
> $$
> \hat{U}_i^t = U_i^t + \frac{\sum\_{j=1}^{N} \mathcal{A}\_{ij} w\_{ij}  U_j^t}{\sum\_{j=1}^{N} \mathcal{A}\_{ij} w\_{ij} }.
> $$
> We denote this variant as MA$^3$S-W and compare the label accuracies (%) of four real-world datasets after applying MA$^3$S-W and MA$^3$S with 200 annotation rounds, and the results are as follows:
> |Dataset|*Income*|*Leaves*|*Music*|*LabelMe*|
> |----|----|----|----|----|
> |MA$^3$S-W|75.00|70.05|**78.13**|**83.67**|
> |MA$^3$S|**75.34**|**72.14**|78.00|82.70|
> ||
>
> Based on these results, we can observe that MA$^3$S-W does not consistently achieve better performance across all datasets as expected. We analyze the possible reasons as follows: on the one hand, the value of $K$ is set to a very small number in MA$^3$S, making the distinction introduced by weighting less pronounced; on the other hand, the process of scaling distances into weights may introduce additional errors. In conclusion, to maintain simplicity, we temporarily refrain from introducing more complex neighbor weighting methods. Thanks again for your valuable comments.
>
> **W3:** The paper currently discusses only single-instance annotations, i.e., selecting one instance per round for labeling, which can be limited in real-world applications. It is recommended to discuss the results of the new strategy under batch annotation, including how the strategy is adapted and how it performs when selecting multiple instances per round.
>
> **Author Response to W3:** Thanks for your valuable comments. Our MA$^3$S can be extended to batch annotation. In the batch setting, we only need to modify Equation (13) to select the top *batch_size* instances with the highest uncertainties. After batch annotation, uncertainties are updated collectively using Equation (5) and then propagated via Equation (12). Due to the reduced number of propagation steps, MA$^3$S will become more efficient. However, since uncertainty cannot be updated and propagated immediately after each individual annotation, its performance may be slightly weaker than in single-instance setting. The reviewer provides a valuable perspective, and we will incorporate the above discussion into the revised version. Thanks again for your valuable comments.

---

> > ### Author Rebuttal · Reviewer_SBcy · 2026-04-02
> >
> > My concerns have been adequately addressed. I maintain the positive score.

---

### Official Review · Reviewer_1V5T · 2026-03-10

**Soundness:** 2
**Presentation:** 2
**Significance:** 2
**Originality:** 2
**Overall Recommendation:** 4
**Confidence:** 3

**Summary:**

This paper proposes MA^3S, a model-agnostic active annotation strategy designed for repeated annotation in crowdsourcing settings. The method aims to reduce annotation redundancy under the assumption that worker reliability is unknown. Experiments on synthetic datasets and four real-world datasets demonstrate improved performance compared with several baselines.

**Compliance With Llm Reviewing Policy:**

Affirmed.

**Final Justification:**

The rebuttal partially addressed my concerns, but the ablation study is limited to a single dataset, and the overall approach still largely combines existing techniques. I raised my score slightly.

**Key Questions For Authors:**

See weaknesses.

**Limitations:**

yes

**Strengths And Weaknesses:**

Strength
1. The paper addresses the practical problem of reducing redundant annotations in crowdsourced labeling.
2. MA^3S combines uncertainty estimation and neighborhood propagation in a conceptually straightforward and easy-to-implement manner.
3. The experiments compare with several baselines and show consistent improvements.

Weakness:
1. The approach mainly combines existing ideas such as margin-style uncertainty estimation and k-NN propagation, leading to modest technical innovation.
2. The theoretical analysis only establishes basic monotonic properties of the uncertainty measure and does not provide stronger guarantees.
3. The paper lacks ablation studies to verify the individual contributions of key components.
4. Equation (10) regarding Euclidean distance appears to contain a notation error.

---

> ### Author Rebuttal · Authors · 2026-03-30
>
> **W1:** The approach mainly combines existing ideas such as margin-style uncertainty estimation and k-NN propagation, leading to modest technical innovation.
>
> **Author Response to W1:** Thanks for your valuable comments. Our main technical innovation lies in improving the repeated annotation strategies, while margin-style uncertainty estimation and k-NN propagation merely serve as the foundation for these improvements. Compared with existing repeated annotation strategies, the proposed MA$^3$S achieves the following key improvements:
> > - **Online procedure.** We leverage a Beta distribution to estimate and update instance uncertainties online, thereby enabling suitability for multi-class scenarios and reducing label redundancy associated with offline procedure. Meanwhile, we also provide theoretical guarantees for this innovation.
> > - **Instance-aware design.** We construct a nearest-neighbor graph to propagate instance uncertainties, fully leveraging instance information in the attribute space and alleviating instance redundancy induced by instance-unaware designs. This innovation avoids repeatedly annotating similar instances.
> > - **Model-agnostic mechanism.** The proposed MA$^3$S can actively select the most valuable instances to annotate based on instance uncertainties, without relying on model feedback. This innovation effectively overcomes the cold-start issue and the need for frequent updates faced by model-centric strategies.
>
> Achieving the above improvements simultaneously represents a significant advancement over existing repeated annotation strategies. Thanks again for your valuable comments.
>
> **W2:** The theoretical analysis only establishes basic monotonic properties of the uncertainty measure and does not provide stronger guarantees.
>
> **Author Response to W2:** Thanks for your valuable comments. For repeated annotation strategies, uncertainty measure is crucial. A desirable strategy is expected to select instances with the highest uncertainty for annotation. Our theoretical analysis shows that the uncertainty measure incorporated in the proposed MA$^3$S is theoretically complete and can distinguish scenarios that existing strategies fail to differentiate. This represents an important advancement and provides a strong theoretical guarantee for the effectiveness of MA$^3$S. Thanks again for your valuable comments.
>
> **W3:** The paper lacks ablation studies to verify the individual contributions of key components.
>
> **Author Response to W3:** Thanks for your valuable comments. MA$^3$S mainly improves repeated annotation performance through uncertainty estimation and uncertainty propagation. In the current manuscript, we have provided separate evidence for their individual contributions: for uncertainty estimation, we offer a rigorous theoretical guarantee via Theorem 3.1; for uncertainty propagation, we demonstrate its effectiveness through the results shown in Fig. 5. To further address the reviewer’s concern, we construct a set of ablation studies. Specifically, we design three ablated variants for MA$^3$S, denoted as MA$^3$S$\_1$, MA$^3$S$\_2$, and MA$^3$S$\_3$. Here, MA$^3$S$\_1$ degrades the uncertainty estimation to be the same as LU, MA$^3$S$\_2$ removes uncertainty propagation, and MA$^3$S$\_3$ combines both modifications (i.e., equivalent to LU). Considering that LU cannot handle multi-class scenarios, we conduct the ablation studies only on the *Income* dataset. We evaluate the label accuracy (\%) of *Income* after applying each variant with 200 annotation rounds, and the results are as follows:
> |Variant|MA$^3$S|MA$^3$S$\_1$|MA$^3$S$\_2$|MA$^3$S$\_3$|
> |----|----|----|----|----|
> |Label accuracy (\%)|**75.34**|74.67|74.83|74.50|
> ||
>
> Based on these results, we can further validate the contribution of each key component in MA$^3$S. It can be observed that removing any component degrades the performance of MA$^3$S. When both components are removed, the worst label accuracy is obtained. We will include the above ablation studies in the revised version. Thanks again for your valuable comments.
>
> **W4:** Equation (10) regarding Euclidean distance appears to contain a notation error.
>
> **Author Response to W4:** Thanks for your valuable comments. In Equation (10), the square should be placed outside the parentheses. We will update Equation (10) as follows:
> $$
> d({x}\_i,{x}\_j) = \sqrt{\sum\_{m=1}^{M}(x\_{im}-x\_{jm})^2}.
> $$
>
> Additionally, we will double-check and improve the writing of our paper. Thanks again for your valuable comments.

---

> > ### Author Rebuttal · Reviewer_1V5T · 2026-04-02
> >
> > My concerns about the limited novelty and the ablation evidence being limited to a single dataset still remain to some extent. However, the rebuttal has partially addressed these concerns, and I am therefore willing to increase my score slightly.

---

### Official Review · Reviewer_nAB7 · 2026-03-14

**Soundness:** 1
**Presentation:** 1
**Significance:** 3
**Originality:** 2
**Overall Recommendation:** 4
**Confidence:** 4

**Summary:**

The work proposes a model-agnostic active learning-based annotation strategy for crowdsourcing. Existing methods are either model centric or do not have instance-awareness in selecting the instances for each annotation round. Here, the authors propose a general beta distribution-based uncertainty estimation and kNN-based instance-aware selection to improve the selection efficiency and final accuracy. Both synthetic and real-data experiments are presented to showcase the effectiveness of the approach.

**Compliance With Llm Reviewing Policy:**

Affirmed.

**Final Justification:**

The rebuttal has resolved my major concerns and I am have also gone through other reviewer's comments and responses. Based on the overall rebuttal, I am raising my score.

**Key Questions For Authors:**

Please see Weaknesses section. In addition, I have a few other questions

1.	In Figure 2, why same color coding is used for categories as well for marking the uncertainties? Also, are you considering label missingness in each rounds of annotation?

2.	Some comments like MA^3S selects more instances from category I_2 than from I_4 etc are not clear from figures 2 and 3. The explanation and the visualization here are quite vague.

3.	In Figure 3, LU seems to have a good selection balance between two classes compared to MA^3S. Why?

4.	In Figure 5, there is no much change in K=0 and K=5. Please clarify.

5.	In real data experiments, 1200 annotation rounds are used. So, are you selecting more than  one instance in each round?

6.	How does the method perform in class-imbalanced scenarios. Will the same algorithm still be optimal?

**Limitations:**

Please discuss the limitations of the approach, especially compared to model-centric approaches and those approaches that automatically learn worker abilities.

**Strengths And Weaknesses:**

Strengths:

1.	The problem is well-motivated, and the introduction section is quite convincing

2.	Reducing redundancy in annotations and selecting most suitable instances through instance-awareness are promising ideas.

Weaknesses:

1.	Authors claim that instance selection does not generally require learning worker abilities, which is not well-substantiated. There exist several works in crowdsourcing that automatically estimates worker abilities and results in improved label integration. There is no mention of such works and potential disadvantages of not considering worker abilities.

2.	The claim for using the beta distribution as in Eq. (2) seems weak. It is mentioned that Ideally workers would reach consensus on the unknown true class and the confusing class…” There can be multiple confusing classes in multi-class scenario.

3.	After Eq. (9) and theoretical analysis, it is mentioned that the proposed strategy can better distinguish {+,-}, and {+,+,+,-,-,-} compared to LU. But this can be simply achieved by selecting the instances that have less labels, right. Hence, U_i^t’s advantage is not well-substantiated.

4.	Case study in Page 4 seems just like a visual illustration of the proposed method’s idea. It does not specify the experiment settings of the case study and did not explain how Figure 1 was generated.

5.	Experiment section is quite weak, and many claims are hard to understand. In synthetic experiments, how are the six categories be assigned to instances are unclear, which also translates to some confusion in understanding Figure 2.

6.	In baselines, it is not mentioned if any model-centric baselines are being used. It is also worthwhile comparing the approach with model-centric baselines and conveying why one should adopt model-agnostic.

---

> ### Author Rebuttal · Authors · 2026-03-29
>
> Thanks a lot for your comments. Please find our detailed responses to your concerns as follows.
>
> **Response to W1:** We fully agree with your view that existing works estimate worker abilities to improve label integration performance. However, it is important to clarify that label integration is a downstream task of repeated annotation we focus on. In label integration, workers have already completed their annotations, providing sufficient labeled data to model their abilities. In contrast, repeated annotation starts from an unlabeled dataset, with workers recruited from an open environment. These conditions make worker ability both unknown and difficult to model due to insufficient labeled data. Therefore, we claim that “**instance selection need not assume that worker abilities are known**.” This is not equivalent to claiming that “instance selection does not generally require learning worker abilities.”
>
> **Response to W2:** We thank the reviewer for raising this interesting question. In fact, when designing Eq. (2), we explicitly considered the case of multiple confusing classes. Our analysis is as follows: if, among multiple confusing classes, the most representative one has a label count close to that of the true class, Eq. (2) can effectively handle this case; if the label counts of the confusing classes differ significantly from that of the true class (i.e., follow a long-tail distribution), they introduce no additional challenges to Eq. (2). Therefore, Eq. (2) remains applicable in multi-class scenarios.
>
> **Response to W3:** Compared with LU, our MA$^3$S can directly distinguish {+,-}, and {+,+,+,-,-,-}, indicating its theoretical completeness. More than that, MA$^3$S offers additional advantages over LU, such as handling multi-class scenarios and being instance-aware.
>
> **Response to W4:** Due to unknown worker abilities and uncontrollable instance uncertainty, the cases shown in Fig. 1 are not easily generated through experiments and are therefore constructed synthetically for illustration.
>
> **Response to W5:** In the synthetic experiments, after generating the instances, we assign them sequentially to the six categories, i.e., the first 30 instances correspond to $\mathcal{I}_1$, instances 30–60 correspond to $\mathcal{I}_2$, and so on. We will provide a more detailed description of the synthetic process in the revised version.
>
> **Response to W6:** For the baselines, MU, LMU, and ALFC used in this paper are all model-centric strategies. We will further emphasize them in the revised version.
>
> **Response to Q1:** The categories do not represent the true class labels of the instances, but rather their multiple noisy label sets. For example, all instances in $\mathcal{I}_2$ have the same label sets {+, −}, and thus share the same initial uncertainty. Therefore, in Fig. 2a, instances within the same category have identical uncertainty (i.e., the same color). Additionally, we do not consider label missingness, as it does not affect repeated annotation. If no worker provides an annotation at round $t$, then round $t+1$ simply serves as the effective $t$-th round.
>
> **Response to Q2:** In the synthetic dataset, instances from $\mathcal{I}_2$ are distributed in the upper half, while those from $\mathcal{I}_4$ are in the lower half, as shown in Fig. 2a. As annotation round increases, instances from $\mathcal{I}_2$ are selected earlier, leading to a faster reduction in their uncertainty, as illustrated in Fig. 2b. In Fig. 4, the curve of $\mathcal{I}_2$ consistently lies above that of $\mathcal{I}_4$, indicating that MA$^3$S prioritizes instances from $\mathcal{I}_2$.
>
> **Response to Q3:** A desirable annotation strategy is expected to prioritize instances with higher uncertainty, regardless of their class. When $T$ ranges from 0 to 10, instances in $\mathcal{I}_2$ (distributed in the upper half of the dataset) exhibit higher uncertainty (when neighborhood information is not considered), and are therefore more likely to be selected. The results shown in Fig. 3 demonstrate that MA$^3$S distinguishes $\mathcal{I}_2$ from $\mathcal{I}_4$ more effectively than LU.
>
> **Response to Q4:** When $K=0$, MA$^3$S is no longer instance-aware and thus only selects instances from $\mathcal{I}_2$ distributed in the upper half of the dataset. When $K=5$, MA$^3$S incorporates neighborhood uncertainty and therefore selects a small number of $\mathcal{I}_4$ instances from the lower half of the dataset, demonstrating its instance-aware capability.
>
> **Response to Q5:** Currently, we select only one instance for annotation in each round. The reviewer provides a valuable perspective, and in future work, we will extend our MA$^3$S to support batch annotation per round.
>
> **Response to Q6:** In theory, class labels do not affect the performance of annotation strategies. They should be driven solely by the uncertainty. Among real-world datasets we use, Leaves is class-imbalanced, and our MA$^3$S still maintains its superiority on Leaves.

---

> > ### Author Rebuttal · Reviewer_nAB7 · 2026-04-06
> >
> > Thanks for addressing my concerns and resolving the gaps. I am raising my score.

---

### Decision · Program_Chairs · 2026-04-30

**Decision:**

Accept (regular)

**Comment:**

This paper proposes MA3S, a model-agnostic active annotation strategy for crowdsourcing. MA3S combines a margin-based uncertainty estimation (via a Beta approximation) with kNN-based uncertainty propagation to enable online, instance-aware, and model-agnostic instance selection. Reviewers generally agree that the problem is important, and that the proposed method is simple, intuitive, and demonstrates consistent improvements over several baselines on both synthetic and real-world datasets.

However, main concerns include limited novelty, relatively weak theoretical analysis, and experimental clarity. The authors’ rebuttal clarifies the method, adds ablations and sensitivity analysis, and resolves most issues. All the reviewers updated their scores positively.

Overall, this is a solid, practical contribution with moderate novelty. I recommend accept.